# Identification and Visualization of Key Topics in Scientific Publications with Transformer-Based Language Models and Document Clustering Methods

**Min-Hsien Weng [1,*]**, **Shaoqun Wu [1] and Mark Dyer [2]**

1   School of Computing and Mathematical Sciences, The University of Waikato, Hamilton 3216, New Zealand
2   School of Engineering, The University of Waikato, Hamilton 3216, New Zealand
*   Correspondence: min-hsien.weng@waikato.ac.nz

**Abstract:** With the rapidly growing number of scientific publications, researchers face an increasing challenge of discovering the current research topics and methodologies in a scientific domain. This paper describes an unsupervised topic detection approach that utilizes the new development of transformer-based GPT-3 (Generative Pretrained Transformer 3) similarity embedding models and modern document clustering techniques. In total, 593 publication abstracts across urban study and machine learning domains were used as a case study to demonstrate the three phases of our approach. The iterative clustering phase uses the GPT-3 embeddings to represent the semantic meaning of abstracts and deploys the HDBSCAN (Hierarchical Density-based Spatial Clustering of Applications with Noise) clustering algorithm along with silhouette scores to group similar abstracts. The keyword extraction phase identifies candidate words from each abstract and selects keywords using the Maximal Marginal Relevance ranking algorithm. The keyword grouping phase produces the keyword groups to represent topics in each abstract cluster, again using GPT-3 embeddings, the HDBSCAN algorithm, and silhouette scores. The results are visualized in a web-based interactive tool that allows users to explore abstract clusters and examine the topics in each cluster through keyword grouping. Our unsupervised topic detection approach does not require labeled datasets for training and has the potential to be used in bibliometric analysis in a large collection of publications.

**Keywords:** topic analysis; language model; document clustering; keyword extraction; bibliometric analysis

## 1. Introduction

With the number of research scientific publications growing rapidly over the past decades, it has become increasingly difficult for researchers to review large volumes of publications to discover the current topics and explore new emerging themes in a scientific field. Topic detection is a process that identifies the main topics among a collection of text data, where each topic is represented by a list of words [1]. LDA (Latent Dirichlet Allocation) and LSA (Latent Semantic Analysis) are popular topic modeling techniques that help to discover, summarize, and represent topics from a large corpus of texts. These methods are commonly used in the analysis of trending topics in social media texts [2]. In recent years, the demand for topic detection tools that can find topics but also help understand how a topic is derived has become increasingly important. Hence, keywords, of up to five or six words, have been commonly used to represent the topics of a document in the literature. They have also served as index terms for searching published literature in bibliographic databases such as Scopus, Web of Science, ScienceDirect, etc. Existing bibliometric text analysis tools, such as SciMAT [3], bibliometrix [4], and VOSviewer [5], use keywords to provide a general idea of the scientific publications under investigation. Most keywords are single words, but two or three words are also common. They are usually assigned by authors based on their own judgment by means of common taxonomy.

However, more recently, the advancement of text mining techniques has automated the keyword extraction process, particularly from a large volume of texts.

Current keyword extraction methods fall into two categories: supervised and unsupervised. The supervised approaches, for example, KEA [5], treat keyword identification as a classification task. It trains a model on top of an external corpus with known keywords to distinguish relevant and irrelevant keywords and the model is then used to predict the keywordness of candidate words. Unsupervised learning approaches such as YAKE! [6], TextRank [7], or RAKE [8] use algorithms to rank candidate words, from which keywords with high ranks are selected. These methods treat the collection of documents as a whole when generating keywords.

There has been research that combines document clustering and keyword algorithms to detect topics, where the documents are split into a number of clusters and topics or keywords are extracted for each cluster. This approach represents a document with a vector of float numbers (also called *an embedding*), which quantifies the semantic meaning of a document, and then generates clusters based on the semantic similarity of documents so that documents with similar meanings are grouped in the same cluster. The most popular clustering algorithms are K-means and HDBSCAN (Hierarchical Density-Based Spatial Clustering of Applications with Noise), where K-means requires a predetermined K, the number of clusters, and HDBSCAN can detect the outliers, the documents not belonging to any clusters. There are a number of document embedding methods, from Bag-of-words (BoW), Word2Vec [9], and Doc2Vec [10] to the most recent, transformed-based such as BERT (Bidirectional Encoder Representations from Transformers) [11] and the GPT-3 similarity embeddings [12]. Radu et al. (2020) [13] and Vahidnia et al. (2021) [14] experimented with Doc2Vec embedding with off-the-shelf clustering algorithms, such as K-means, hierarchical agglomerative clustering, and deep embedded clustering [15], on publication abstracts and then used the TF-IDF terms to label each cluster. Their results showed that the use of Doc2Vec embedding improves the accuracy of clustering algorithms.

However, there is limited research in the area of topic analysis that takes advantage of the transformer-based language models such as BERT and GPT-3 to represent the semantic meaning of the text. Our main contribution is to demonstrate that a simple method, document clustering algorithms combined with the transformer-based text representation, can be used to group texts at different levels: individual words, multiple words, sentences, or a document, by their semantic meanings. Furthermore, the visualization techniques that are commonly used to present document clustering results are used to illustrate the natural grouping of documents and topics, which adds a new angle to interpreting topics through a list of keywords. Our approach is illustrated using a small collection of 593 publication abstracts gathered from Scopus using the combination of terms *machine learning* and *urban study*. A web-based interface is developed to showcase our approach, which has the potential to be used in any text analysis tools that need to provide a fine-grained topic analysis of a large collection of publications.

## 2. Background and Relevant Literature

This section provides an overview of the techniques used in this study, particularly document clustering algorithms, pretrained embedding models and their applications in document clustering, and keyword extraction.

### 2.1. Document Clustering Algorithms

Document clustering techniques discover the natural groupings among a collection of documents based on text similarity. They are unsupervised methods that do not require prior document labeling, where a document is represented by a high dimensional semantic vector space and similar documents tend to stay close and form a dense area. Used as an alternative to text classification techniques, they are suitable for detecting new emerging topics that did not occur before [16]. The K-means and HDBSCAN (Hierarchical Density-Based Spatial Clustering of Application with Noise) algorithms are the two most used

clustering algorithms. K-means attempts to partition the documents into a set of k clusters, where k is a pre-defined number decided by the user. This algorithm uses an iterative procedure to assign every document to its closest cluster whilst updating the cluster centroids. However, the number of clusters k is hard to determine unless users have prior knowledge and a deep understanding of their data to evaluate the clustering results and find the correct clusters [17], and the algorithm is not sensitive to noise data, i.e., the documents not belonging to any of the clusters. HDBSCAN [18] produces clusters of data points (in the form of vectors) that have a higher density in the neighboring regions that contain at least a minimal number of data points. HDBSCAN uses GLOSH (Global-Local Outlier Scores from Hierarchies) to detect outliers that are decided by a score (from 0 to 1) when the density of outlier points is much lower than their closest cluster that they are most associated with [19]. Therefore, compared to K-means, HDBSCAN can discover clusters with variable density and different shapes, and achieve better clustering results [20] due to its outlier detection.

### 2.2. Pretrained Language Models and Applications

Natural language processing applications such as text classification, document clustering, machine translation, and machine understanding rely on numerical representation of texts (vectors of float numbers or embeddings) that can be mathematically computed. Bag-of-words is the simplest representation that treats each word in a document as an atomic indicator and uses word counts to produce sparse vectors; however, it fails to recognize the words with semantic similarity, e.g., car and bus, because word counts are not sufficient to encode the meaning of a word [21]. Word2Vec embedding [9] is an advancement that represents the word with a dense vector of a few hundred dimensions by utilizing the information of neighboring words. It represents similar words (car and bus) with similar embeddings that capture their semantic and syntactic relations. Doc2Vec [10] is a document representation of Word2Vec that takes the word order into account. Comparative studies by Radu et al. (2020) [13] and Vahidnia et al. (2021) [14] have shown that Doc2Vec models with off-the-shelf clustering algorithms such as K-means and DBSCAN [22] and deep embedded clustering [15] improve the accuracy of document clustering on scientific publications and outperform classical bag-of-words. However, for Word2Vec and Doc2Vec embeddings, only one vector is generated for a word, which fails to embed different senses of a word, for example, the word bank regardless of whether it is used in *river bank* and *commercial bank*.

The latest technique, transformer-based embedding such as BERT (Bi-directional Encoder Representation from Transformer) [11], considers the surrounding context of a word and produces different vectors of a word depending on its context. The first BERT model was trained on BooksCorpus (800 million words) [23] and English Wikipedia (2500 million words) and on two tasks: predicting the masked words, and predicting whether the next sentence follows the first sentence given a pair of sentences. Pretrained BERT models, trained with slightly different neural network architectures or on different datasets, have been widely used in text mining tasks that require vector representation of the text such as detecting cyberbullying contents on social networks [24], recommending news articles relevant to reader interests [25], searching scientific articles with specific criteria [26], and detecting credible disaster information from massive social media posts [27]. However, BERT models are restricted to short texts of a few hundreds of tokens (from 384 to 512 tokens) [28] due to the transformer's quadratic computational complexity with the input length. There are only a few studies that have utilized pretrained BERT models for document representation in document clustering. A topic analysis by [29] combined a BERT model and k-means algorithm detects the topics and clusters from a collection of geospatial research publications. Comparative studies by Altuncu et al. (2021) [30] and Nasim and Haider (2022) [31] reported that the use of BERT models improved the performance of document clustering algorithms such as K-means, DBSCAN, and Affinity Propagation [32].

Recently, a number of transformer-based language models have been developed to learn the embeddings for texts. The sentence-BERT model [33], a modification of BERT, in a

study by Li et al. (2020) [34] shows that pretrained BERT models without further fine-tuning perform poorly in semantic similarity tasks such as document clustering. Sentence-BERT utilizes the Siamese and triple network structures to fine-tune pretrained BERT models to derive semantically meaningful context vectors, where the text similarity can be estimated using the cosine similarity. A study by Ito and Chakraborty (2020) [35] uses the sentence-BERT model and the K-means algorithm to cluster COVID-19-relevant tweets and discover the topics that reveal useful information to assist government decision-making.

In parallel with BERT models, GPTs (Generative Pretrained Transformers) such as GPT-2 [36] and GPT-3 [37] are language models that are trained on massive amounts of texts (400 billion words) to generate realistic human text. GPT-3 offers a set of models for NLP applications that require an understanding of the content such as language translation, classification, text sentiment, and summarization. Recently, three families of embedding models were made available for different functionalities: text search, text similarity, and code search. These embedding models can be used directly to generate embeddings that contain an information-dense representation of the semantic meaning of a piece of text. In this study, we use pre-trained similarity embedding models for clustering. Similar to other GPT-3 models, the similarity embedding provides four models that all are trained on the same dataset with different parameter sizes: Davinci (175 billion), Curie (6.7 billion), Baggage (1.3 billion), and Ada (350 million). In this study, as suggested by Neelakantan et al. (2022) [12], the Curie model is chosen for its better performance than Ada or Baggage and a close performance to the most capable Davinci.

### 2.3. Keyword Extraction

Here, we refer to a keyword as a term that constitutes a single word (e.g., *temperature*) or multiple words (e.g., *cooling systems*). When publishing, authors are often required to assign up to five or six keywords to the research article. These keywords tend to be general and mainly for catalog purposes. Keyword extraction techniques have been developed to automatically identify a list of terms that provide a more comprehensive summary of a document, and it is a technique widely used in information retrieval and topic modeling. The common approaches [38] to derive keywords include:

- Rule-based linguistic approaches that use linguistic knowledge and rules;
- Statistical approaches (such as TF-IDF) that use term frequency and co-occurrence statistics;
- Domain knowledge approaches that utilize an ontology in a domain to identify keywords;
- Machine learning approaches that use algorithms to automatically detect keywords.

Hence, our literature reviews focus on machine learning approaches that are used in this study. The machine learning approaches are of two types: supervised and unsupervised. Supervised approaches train models using human-labeled keywords. The KEA (Keyphrase Extraction Algorithm) model [5] is one of the most used and was built using author-assigned keywords to identify relevant and non-relevant candidate keywords. However, the supervised approach requires human-labeled data, which is not always available [39]. There are publicly accessible topic datasets labeled with keywords in the areas of public health, biomedicine, and pandemic impact, provided by research analytics tools such as the SciVal service [40], but they are not particularly useful for this study, which focuses on different domains (e.g., machine learning and urban planning).

In contrast, unsupervised approaches use rank algorithms to select the representative keywords from a list of candidates. For example, TextRank [7] uses a word graph to represent a document, where the nodes are nouns or verbs and their occurrences within a window (2 to 10 words) are edges. Keywords are selected using Google's PageRank algorithm [41]. Rake [8] (Rapid Automatic Keyword Extraction) uses the word statistics (word frequency and word co-occurrences) of a document to determine the keywords from a list of content words that convey the meanings and exclude punctuations and function words such as and, of, and the. YAKE! [6] calculates and ranks candidates (one or two

words) by a combined score of word statistical features, such as word case (uppercase is more important), position, frequency, relatedness to context, and occurrences in different sentences.

Nowadays, unsupervised algorithms are the most commonly used because they can be applied to the texts in different domains and languages, especially when human annotated labels are not available. However, they tend to generate similar candidate keywords such as *deep learning analytics*, *learning technique*, and *machine learning* that express a similar concept and therefore reduce the informativeness and representativeness of the keyword list [42]. In this study, we incorporate the Maximal Marginal Relevance (MMR) [43] ranking algorithm to select candidate keywords and remove redundant candidates that have similar meanings from the keyword list. The MMR algorithm has been adopted in information retrieval and recommendation systems to provide relevant and interesting results that users have preferences over [44].

## 3. Methodology

Here, we attempt to provide a web-based topic analysis tool that allows researchers to explore the current topics in a research area and discover the methods and data that are deployed in the investigation. Urban planning is the study of interest in this paper as our team has received funding (New Zealand National Scientific Challenge for Better Built Homes, Towns and Cities) for research in this area and published several publications [45–48]. In this study, we focus on research articles in the area of using AI and machine learning in urban governance/design/planning/study with urban data. The abstracts of 593 articles were collected from Scopus using the search term ("*AI*" OR "*machine learning*") AND ("*urban governance*" OR "*urban design*" OR "*urban planning*" OR "*urban data*" OR "*urban studies*"). Along with the metadata such as titles, citations, and author list, the abstracts were downloaded from Scopus (accessed on 10 February 2022) and served as our study data. The methods used in this study comprised three phases, an overview of which is illustrated in Figure 1.

1.　The **iterative clustering phase** represents abstracts as GPT-3 similarity embeddings and uses the HDBSCAN algorithm along with silhouette scores [49] to divide abstracts into clusters.
2.　The **keyword extraction phase** identifies candidate words from each abstract and selects 5 keywords from candidate words with the MMR ranking algorithm to represent each abstract in the abstract clusters.
3.　The **keyword grouping phase** represents keywords as GPT-3 similarity embeddings and uses the HDBSCAN algorithm and silhouette scores again to form keyword groups to represent topics in an abstract cluster.

The abstract clustering and keywords grouping results are presented in a web-based interactive visualization tool, where the user can inspect the topics of abstract clusters. The visualization will be covered in detail in the result section.

### 3.1. Iterative Clustering

This phase divides abstracts into small clusters iteratively according to the semantic similarity through the three steps shown in Figure 2. In Step 1, we use the pretrained GPT-3 similarity embedding model ('*text-similarity-curie-001*') through OpenAI's API service to map the abstracts to the embeddings with 4096 semantic features. In Step 2, the high-dimension space of the abstract embeddings is reduced to a reasonable range because HDBSCAN requires the dimension size to be smaller than the number of abstracts, in our case, smaller than 593 (the total number of abstracts in our dataset). This also minimizes the computation complexity, resulting in an increased clustering accuracy. We use a dimension reduction technique, Uniform Manifold Approximation and Projection (UMAP) [50], along with the HDBSCAN algorithm to determine a range of dimensions that lead to a result that contains more than one distinct cluster. The experiments with a number of different sizes show that a dimension size below 500 always produces at least 3 or more clusters.

Therefore, the embedding dimension is reduced to a range of 20 to 500 when tuning the HDBSCAN parameters in Step 3.

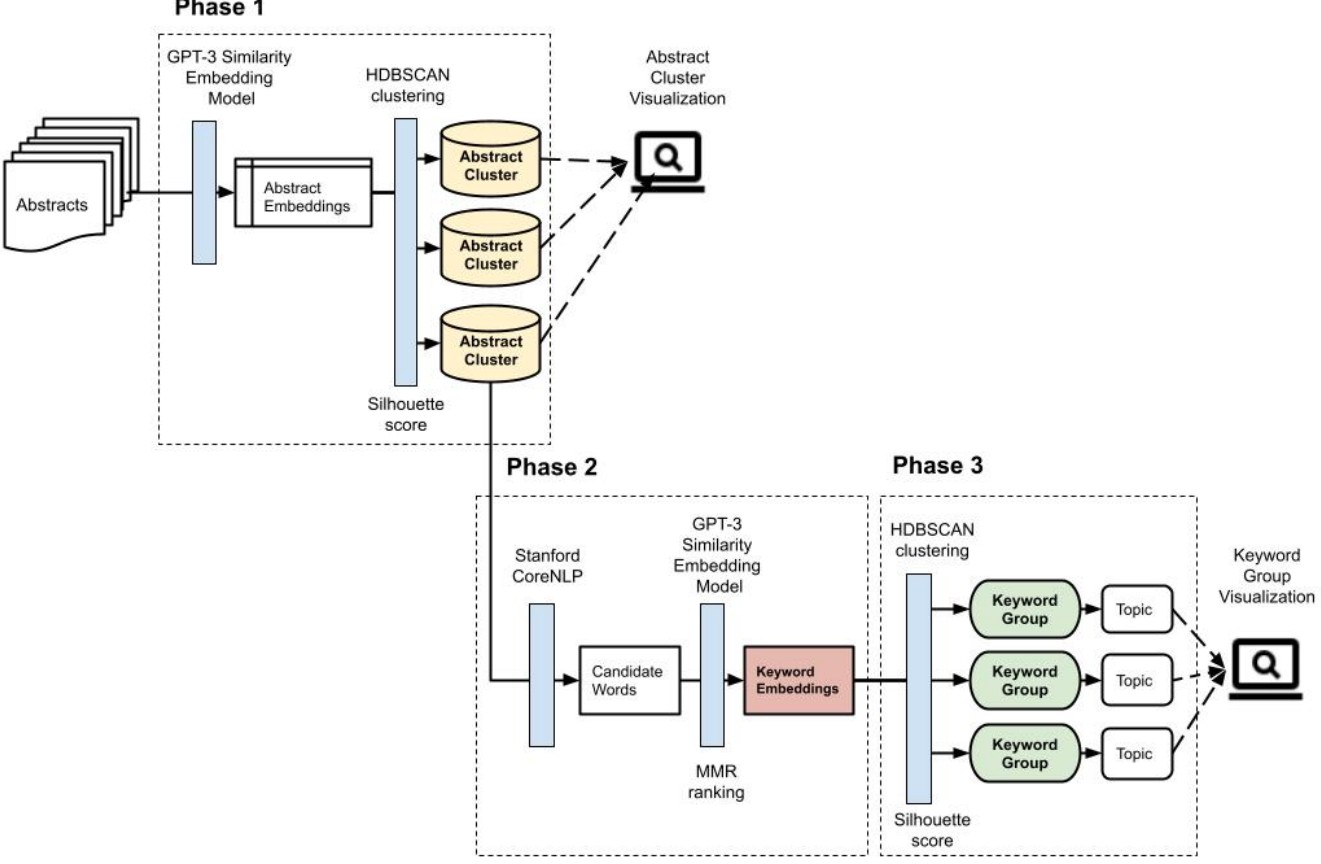

**Figure 1.** An overview of three phases: iterative clustering, keywords extraction, and keywords grouping.

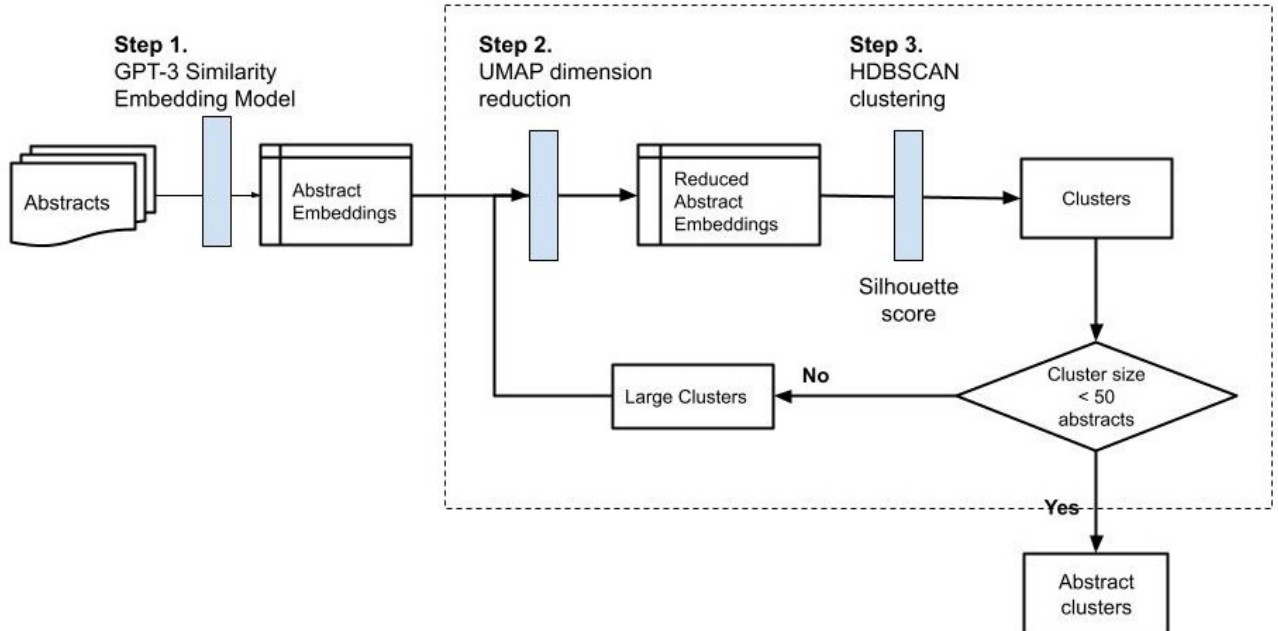

**Figure 2.** The iterative clustering procedure.

In Step 3, HDBSCAN is used to separate the abstracts into small clusters through several iterations as shown in Figure 2. Each iteration produces a number of clusters of different sizes and outliers are treated as one cluster and used in the next iteration because some of them are data points that fall on the cluster borders. The cluster that contains more than 50 abstracts is fed to the next iteration. This process continues until the number of abstracts in every cluster is fewer than 50, a stopping threshold that we chose for good usability when visualizing clustering results and it can be adjusted to 40 or 60. The larger the threshold is, the fewer the number of clusters generated.

HDBSCAN has a number of parameters for fine-tuning the clustering results. We focus on the *minimal cluster size* parameter because it plays an important role in selecting the clusters from a number of candidate clusters whilst other parameters are used as supplementary when the *minimal cluster size* alone cannot make decisions [51]. A larger minimal cluster size results in a fewer number of clusters as small clusters are merged to a large cluster at the cost of a reduced clustering quality. On the other hand, a small minimal cluster size leads to a large number of micro clusters (2 to 4 documents per cluster) and outliers. Therefore, the experiments were conducted by adjusting two parameters: the dimension size of abstract embeddings from 20 to 500 (i.e., 500, 450, 400, 350, 300, 250, 200, 150, 100, 95, 90, 85, 80, 75, 70, 65, 60, 55, 50, 45, 40, 35, 30, 25, 20) and the minimal cluster size from 10 to 50, in increments of 5. In total, there are 225 ($25 \times 9$) combinations of parameters, and each combination produces a set of clusters, which is then evaluated with their silhouette scores to determine a set of optimal clusters.

The silhouette score of a cluster ($C_i$) is calculated using the formula $\frac{b-a}{\max(a,\,b)}$, where $a$ is the *intra-cluster similarity*, which measures the mean similarity of an abstract to all the other abstracts within the same cluster; and $b$ is the *outra-cluster similarity*, which measures the mean similarity of an abstract to all abstracts within the nearest cluster, which has the smallest mean dis-similarity to the cluster $C_i$. The final score is normalized to a decimal value between 1 and $-1$, where a large positive score indicates the abstracts within a cluster are highly similar to each other. Studies by Arbelaitz et al. (2013) [52] and Rendón et al. (2011) [53] suggested that the silhouette score is one of the best clustering validation indexes.

### 3.2. Keywords Extraction

This phase identifies a number of keywords to represent an abstract in each abstract cluster. Keyword extraction is commonly used to generate a summary of a text or to build indexes for browsing a collection of documents. In this study, we define the keywords as the descriptive terms that capture the meaning of an abstract and they are nouns and compound nouns made of two to six words.

Figure 3 illustrates the three-step procedure. Step 1 identifies a list of candidate words from an abstract. The Stanford CoreNLP toolkit [54] splits an abstract text into individual sentences and tokenizes each sentence to words that are then lowercase and lemmatized (e.g., "*Studies*" to "*study*"). The Stanford POS tagger parses a sentence and assigns a Part of Speech (POS) tag to individual words according to their syntactic role, e.g., "adjective", "noun", or "verb". Compound nouns are identified by matching the POS tags of words in a sentence against a set of predefined syntactic patterns as shown in Table 1, where NN stands for a noun and JJ an adjective, and bracket [] means optional. For instance, in the tagged sentence:

*We_PRP integrate_VBP* **visualization_NN techniques_NNS** *with_IN* **machine_NN learning_NN models_NNS** *to_TO facilitate_VB the_DT detection_NN of_IN* **street_NN view_NN patterns_NNS**

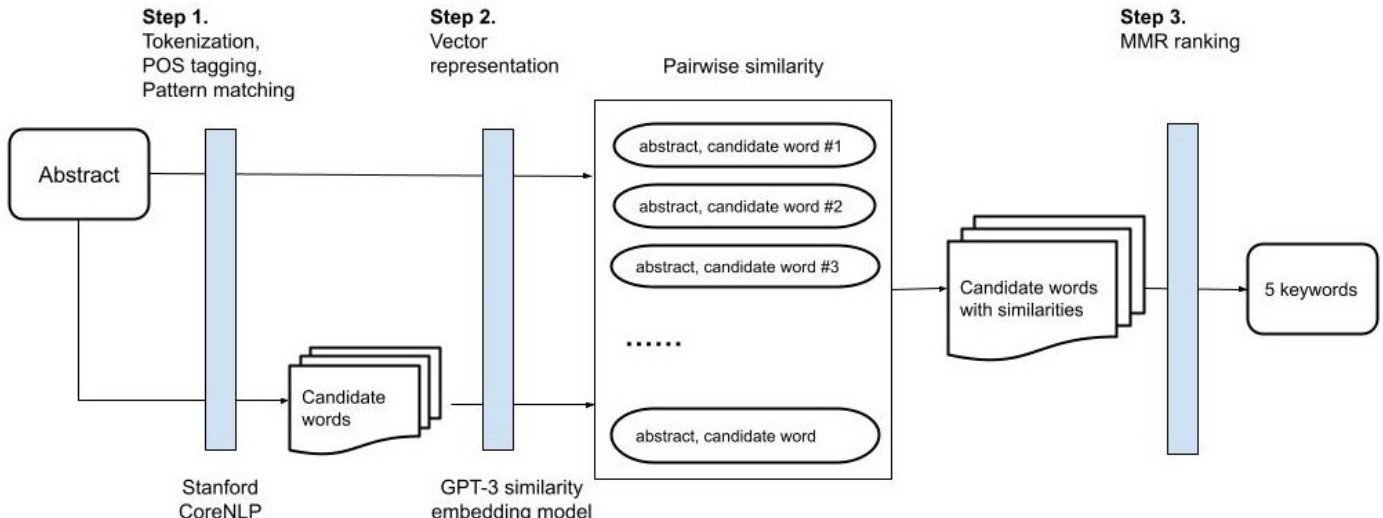

**Figure 3.** The keyword extraction procedure.

**Table 1.** Syntactic patterns for candidate words and examples.

| Pattern | Description | Examples |
|---|---|---|
| NNs | two or more nouns | *land parcel, planning practice, machine learning approach, land surface temperature* |
| JJ + NNs | one adjective plus one or more nouns. | *important variable, main contribution, urban heat island, urban heat island research* |
| JJ + [and] + JJ + NNs | one adjective plus an optional conjunction *and* plus one adjective plus one or more nouns. | *relevant spatial scale, physical and socioeconomic characteristic, major urban environmental problem* |
| [JJ] + NN + and + NNs | an optional adjective plus a noun plus the conjunction *and* plus one or more nouns. | *air and noise pollution, urban greenery and planning practice, urban planning, and land use management* |

The three identified candidate words are *visualization technique*, *machine learning model*, and *street view pattern* as they all follow the pattern NNs (two or more nouns) in Table 1.

Step 2 measures the similarities between a candidate word and the abstract it comes from to determine the representativeness. The candidate words and abstracts are converted into the GPT-3 embeddings. The pairwise cosine similarity is calculated for each pair of candidate words and abstracts. Then, the candidate words are ranked by their similarity to the abstract. Step 3 selects five candidate words using the Maximal Marginal Relevance (MMR) algorithm [43] to form the list of keywords of an abstract. MMR selects keywords that are highly similar to the abstract but have a low similarity to other keywords so that the keyword list contains fewer keywords that have similar meanings. Taking the abstract shown in Figure 4 as an example, the top five candidate words ranked by similarity to the abstract are *urban heat island research, urban heat island study, urban heat island phenomenon*, and *urban heat island*, where the word *urban heat island* repeatedly appears in all five candidate words. Using MMR, *urban heat island research, land parcel, machine learning approach, important variable, physical and socioeconomic characteristic* are chosen as the final five keywords to represent the abstract. Figure 4 also includes the keywords that authors provide. This result shows a certain overlapping (*urban heat island effect, machine learning*, and *socioeconomic vulnerability*) between author keywords and auto generated keywords. The differences

indicate that author keywords do not necessarily come from abstract texts, for example, *variable selection* and *biophysical vulnerability* do not occur in the abstract, and *biophysical vulnerability* is not even in the content of the publication.

**Abstract:** Despite the urban heat islands phenomenon has long been recognized as a major urban environmental problem, it was not until recently that this urban phenomenon gained attention from the discipline of urban planning. To integrate the findings of the **urban heat islands research** into the planning practice, the relationship between land surface temperatures and urban physical and socioeconomic characteristics should be addressed at the planning relevant spatial scale, a **land parcel**. Using a parcel as a unit of analysis, this study proposed to use a **machine learning approach** to identify **important variables** in the formation of urban heat islands in Indianapolis, Indiana. Applying random forest method to planning zones, this study identified planning zone specific urban **physical and socioeconomic characteristics** that are important for the interpretation of urban heat islands phenomenon of Indianapolis, Indiana. The main contribution of this study is twofold: to integrate urban physical and socioeconomic characteristics into a **land parcel** for the better interpretation of the result of urban heat islands study into planning practice and to apply machine learning approach to identify highly determinant variables in the formation of urban heat islands.

**Author Keywords:** biophysical vulnerability; machine learning; random forest; socioeconomic vulnerability; urban heat island effect; variable selection

**Figure 4.** An example of an article abstract [55] highlights our extracted keywords in the text and lists the author-assigned keywords under the text.

### 3.3. Keywords Grouping

This phase uses the same technique, which is inspired by BERT-Topic [56] and Key-BERT [57], as that of the iterative abstract clustering to group the keywords of abstracts to reveal a number of underlying topics in an abstract cluster. Here, we only briefly describe the procedure as the technical details are discussed in Section 3.1. First, the keywords are collected from abstracts within an abstract cluster and converted to GPT-3 embeddings. Second, UMAP is used to reduce the dimension size of embeddings. Third, HDBSCAN and silhouette scores are used to divide the keywords into groups and determine the optimal keyword groups. Unlike abstract clustering, only one iteration is carried out and the outliers are discarded because we are only interested in those that are suitable for representing the topics in an abstract cluster.

## 4. Experiment Results and Visualization

To illustrate the capability of these methods, this section presents the results of each phase described in Sections 3.1 and 3.3 in a web-based visualization tool we developed in this project.

### 4.1. Abstract Clustering Results and Visualization

Four clustering iterations are conducted in the iterative clustering phase. An iteration stops when a cluster contains fewer than 50 abstracts. In total, 24 clusters formed from 593 abstracts, each with 10 to 46 abstracts. Table 2 shows the number of abstracts and silhouette scores of each cluster in each iteration. The first iteration generates 5 small clusters (#1 to #5) with fewer than 50 abstracts; 2 large clusters with 62 and 107 abstracts, respectively; and 287 outliers, which have a lower density indicated by the negative silhouette score ($-0.76$). In the second iteration, the 2 large clusters (62 and 107) are split into 6 subclusters (#6 to #11) and the 287 outliers from the first iteration are divided into 5 subclusters (#12 to #16). From 187 outliers in the second iteration, 5 subclusters (#17 to #21) are generated in the third iteration and 106 outliers are fed to the fourth iteration. In the fourth iteration, three more subclusters (#22 to #24) are formed.

**Table 2.** The results of four iterations of abstract clustering. Each abstract cluster is labeled with a sequence number #1, #2, and #3, etc.

| | 1st Iteration | | 2nd Iteration | | 3rd Iteration | | 4th Iteration | |
|---|---|---|---|---|---|---|---|---|
| | Number of Abstracts | Silhouette Score | Number of Abstracts | Silhouette Score | Number of Abstracts | Silhouette Score | Number of Abstracts | Silhouette Score |
| | (#1) 16 | 0.99 | | | | | | |
| | (#2) 21 | 0.88 | | | | | | |
| | (#3) 38 | 0.74 | | | | | | |
| | (#4) 31 | 0.69 | | | | | | |
| | (#5) 31 | 0.51 | | | | | | |
| | 62 | 0.77 | (#6) 23 | 0.85 | | | | |
| | | | (#7) 25 | 0.67 | | | | |
| | | | (#8) 14 | −0.14 | | | | |
| | 107 | 0.71 | (#9) 33 | 0.87 | | | | |
| | | | (#10) 42 | 0.84 | | | | |
| | | | (#11) 32 | −0.41 | | | | |
| 593 | | | (#12) 13 | 0.95 | | | | |
| | | | (#13) 12 | 0.82 | | | | |
| | | | (#14) 23 | 0.66 | | | | |
| | | | (#15) 17 | 0.54 | | | | |
| | | | (#16) 35 | 0.16 | | | | |
| | | | 187 | −0.50 | | | | |
| | 287 | −0.76 | | | (#17) 10 | 0.74 | | |
| | | | | | (#18) 15 | 0.71 | | |
| | | | | | (#19) 25 | 0.61 | | |
| | | | | | (#20) 12 | 0.60 | | |
| | | | | | (#21) 19 | 0.36 | | |
| | | | | | 106 | −0.48 | (#22) 26 | 0.33 |
| | | | | | | | (#23) 34 | 0.31 |
| | | | | | | | (#24) 46 | 0.02 |

The silhouette scores in Table 2 reflect the quality of the clusters in each iteration. A cluster with a positive score close to 1 suggests that the abstracts within this cluster are highly similar to each other. By contrast, the outliers are typically associated with a negative score and each iteration always produces a number of outliers. We will discuss the reason for not discarding the outliner in the next section. In general, 24 abstract clusters appear to be well defined in 3 bands: 17 clusters (65% of abstracts) with good scores above 0.5, 5 clusters (27% of abstracts) with medium scores between 0.5 and 0, and 2 clusters (8% of abstracts) with negative scores. On average, the abstract cluster has a score of 0.55.

To visualize the clustering results, the dimension of an abstract embedding, produced by GPT-3 Similarity Embedding Model, is reduced using UMAP. The reduced embedding is then projected to the position (x, y) on a plane of two dimensions, one for the x-axis and one for the y-axis. The visualization uses the Plotly JavaScript open-source graphing library (https://plotly.com/javascript/, accessed on 1 January 2022) to draw a scatter plot, with each abstract represented as a dot that corresponds to its projected position (x, y). The dot color encodes the clustering membership of each abstract in the first iteration. The blue dots are of the five clusters #1 to # 5 in Table 2; the green dots the cluster #6, #7, and #8; and

the orange dots the cluster #9, #10, and #11. The 287 outliers from the first iteration are colored in gray. It is interesting that these outliers are clustered in three areas, where the left region is the cluster #12 to 16#, the middle is cluster #22 to #24, and the right is cluster #17 to #21.

Clicking on a cluster's color region on the scatter plot, say blue, displays seven subsequent abstract clusters (the right panel in Figure 5). These clusters are represented in a list, where they are formatted using the JavaScript open-source Pagination (https://pagination.js.org/index.html, accessed on 1 January 2022) and CSS Bootstrap (https://getbootstrap.com/, accessed on 1 January 2022) libraries. Each abstract cluster shows its silhouette scores and a list of representative terms to provide an overview of this cluster. Terms (two-word compound nouns) are extracted from the abstracts in a cluster and ranked by a score summing up the frequency and range, where the frequency is the total number of occurrences of a term and the range is the total number of abstracts that a term appears in for the cluster. We focus on compound nouns of two words because nouns are content words that convey the meaning and compound nouns carry more specific meaning than single nouns. The top of the right panel in Figure 5 shows nine common terms in the blue cluster and some of these terms, such as *machine learning*, *urban data*, *urban design*, and *urban planning*, are the terms that we used to search the abstracts in Scopus. Beneath are the five clusters, sorted by Silhouette score, along with the number of abstracts and the frequent terms of each cluster. The frequent terms allow further examination of the content of a cluster. For example, the cluster #1 is about *urban air quality* and relevant technologies such as management system, information system, and control system; the cluster #2 *urban heat* and *air temperature*; the cluster #3 *urban environment* and *human perception*; the cluster #4 *urban land use* and *urban growth;* and the last cluster #5 contains a diversity of terms that do not appear in other clusters such as *urban governance*, *urban system*, and *urban form*.

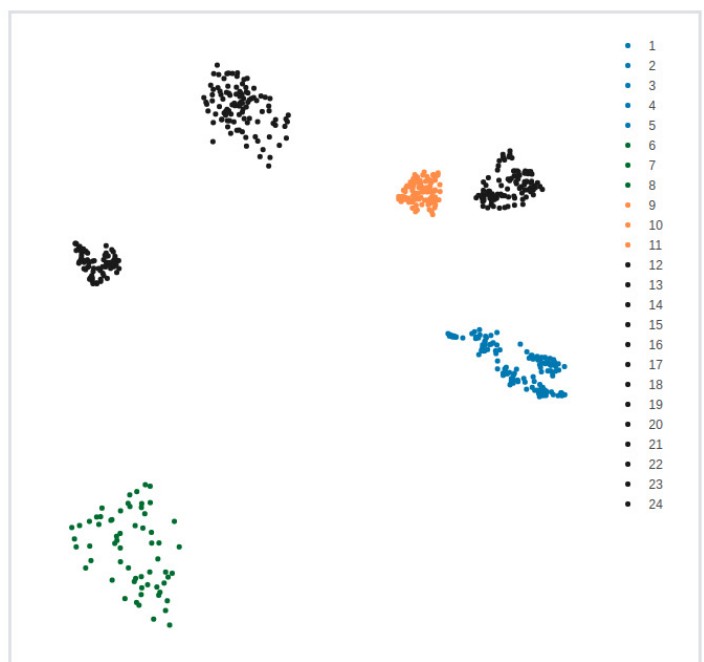

**Figure 5.** Abstract cluster visualization for all clusters from the iterative clustering phase. On the left, the scatter chart of abstract vectors reveals three color regions (blue, orange, and green) along with three gray regions for outliers. Clicking on the blue region retrieves and displays the abstract clusters in the right panel.

Clicking the cluster #2 in Figure 5 displays its total number of abstracts and its frequent terms, as shown in Figure 6. The top panel lists the top 10 frequent terms in a cluster, and the bottom panel displays the details of the abstracts. The number (12) in brackets following

a term, e.g., *surface temperature* (12), is the number of abstracts containing the term. Clicking a term, say *temperature data*, displays the abstracts containing the term in the bottom panel. Each abstract highlights the selected term along with the auto-generated keywords in the keyword extraction phase. More information about an abstract, including article titles, authors, author keywords, citations, publication year, and DOI, are given at the end. This result shows a very limited number of overlapping terms in the cluster. Only four terms (*surface temperature*, *urban heat*, *heat island*, and *land surface*) occur in more than 10 abstracts (half of the cluster) whilst most of the terms occur only once. The diverse terms in this cluster make it difficult to find common terms representing the topics.

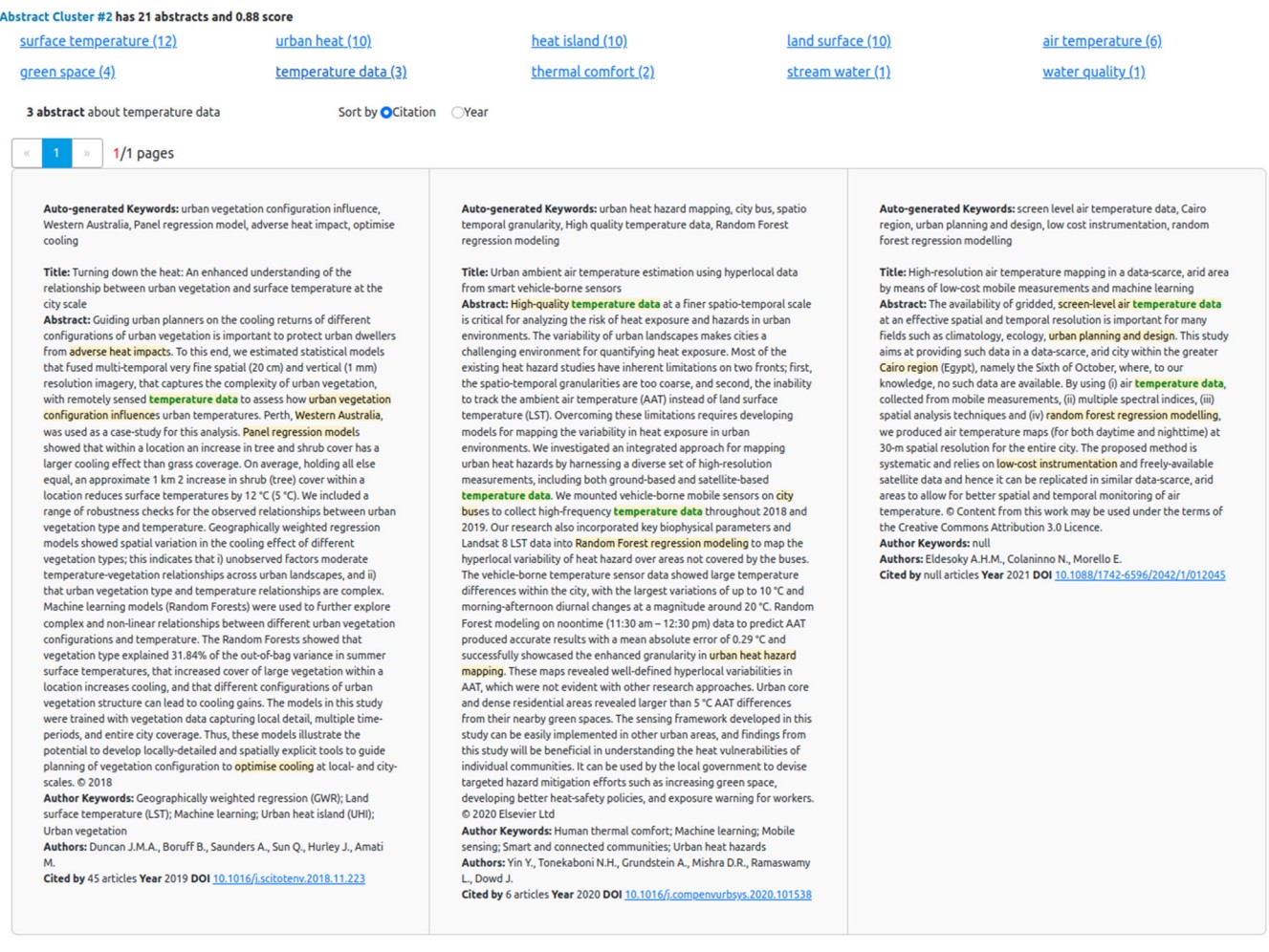

**Figure 6.** Abstract cluster visualization for a single cluster, #2. The top panel displays 10 frequent terms of the selected cluster. Clicking a term, *temperature data,* displays the abstracts and the auto-generated keywords from the keyword extraction phase.

*4.2. Keyword Grouping Results and Visualization*

In the keyword extraction phase, 2965 keywords are collected from 593 abstracts, with 5 keywords from each abstract. To be noted, there are duplicates in keywords as one or more abstracts can contain the same keyword. The keyword grouping phase generates 98 keyword groups from 24 abstract clusters, with an average of 4 groups per cluster. This means that an abstract cluster may contain more than one topic, with a keyword group indicating a topic. Table 3 shows the statistics of keyword groups in individual clusters, including the abstract cluster sequence number, number of abstracts, number of keywords, and coverage of the keyword group. Silhouette scores are used to evaluate and validate the quality of the keyword grouping results, where the keyword groups in a cluster are ordered by silhouette scores. For simplification, the keyword groups with a negative silhouette

score are not listed in the table as these keyword groups contain a set of keywords classified as outliers that do not form a topic. The results show that 8 out of 24 clusters contain 2 keyword groups, 7 clusters have 3 groups, 7 clusters have 4 groups, and 2 clusters have 5 groups. In total, there are 75 keyword groups across all the clusters and each cluster contains 3 keyword groups on average.

The coverage is used to evaluate the representation of a keyword group in a cluster. It is calculated as the ratio of the abstract number of a keyword group to the total number of abstracts of a cluster. A greater coverage indicates a higher percentage of keywords coming from different abstracts in the cluster and provides a greater representation. For example, a keyword group with 100% coverage contains at least one keyword from every abstract of the cluster. The result in Table 3 shows that all abstract clusters have one or more keyword groups with good coverage (>50%) and 7 clusters (#4, #8, #12, #15, #17, #18, #20) have at least one keyword group with nearly full coverage (>90%). On average, a keyword group has a coverage of 68%, ranging from 45% to 96%, which indicates our keyword groups have good representation for the topics in the cluster.

Figure 7 shows the keyword group visualization interface, which allows the user to select and examine the keyword groups in each cluster. The visualization has two components: the left panel presents a scatter chart of keyword groups, and the right panel lists the silhouette scores and keywords in each group. In the left panel, the user selects a cluster #2 from the dropdown menu to retrieve the keyword groups. The left panel, as illustrated in Figure 7a, contains a scatter chart of four keyword groups in blue, orange, green, and red, respectively, where a dot represents a keyword embedding in two dimensions after being projected to its x and y value on the x-axis and y-axis using UMAP. The chart shows that each keyword group forms its own area, well separated from each other, with only a few overlapping dots. The area densities (from dense to sparse) of the three keyword groups reflect their silhouette scores from 0.94 (group #1 in blue) to 0.47 (group #4 in red). The right panel displays the keyword groups in detail, including a list of keyword examples, with more keywords revealed when the plus sign at the right corner is clicked. It shows that group #1 is about machine learning, group 2 place names, group 3 temperatures and weather datasets, and group 4 urban heat.

Clicking a keyword group number, say #4, displays the visualization interface of an individual keyword group, as shown in Figure 7b, where all the keywords in the group are listed in the top panel. In this case, *urban heat island*, *heat impact*, and *urban thermal environment* is the common theme (or topic) revealed by this keyword group. Clicking on a keyword, say *urban heat island*, displays three abstracts in the bottom panel that contain the clicked keyword highlighted in orange. These results show keywords in this group have a low amount of overlapping terms, and most keywords share a certain similarity (such as *adverse heat impact*, *urban heat hazard*, and *urban thermal environment*) but contain different words.

The total processing time on our dataset (593 abstracts) is about 1.5 h, with each phase roughly 25 to 30 min on a 6-core computer with 16 GB memory. The software implementation is listed in Supplementary Material and described in Appendix A.

**Table 3.** Statistics of keyword groups in abstract clusters: the number of abstracts, number of keywords, and coverages. Note: the sequence numbers in the first column reference the abstract sequence numbers in Table 2.

| Abstract Cluster Sequence Number | Number of Abstracts | Group 1 | | Group 2 | | Group 3 | | Group 4 | | Group 5 | | Average Coverage |
|---|---|---|---|---|---|---|---|---|---|---|---|---|
| | | Number of Keywords | Coverage | Number of Keywords | Coverage | Number of Keywords | Coverage | Number of Keywords | Coverage | Number of Keywords | Coverage | |
| 1 | 16 | 11 | 69% | 9 | 56% | 9 | 56% | | | | | 60% |
| 2 | 21 | 16 | 67% | 10 | 43% | 15 | 62% | 21 | 86% | | | 65% |
| 3 | 38 | 35 | 84% | 14 | 34% | 24 | 66% | 14 | 37% | 10 | 26% | 49% |
| **4** | **31** | **18** | **55%** | **131** | **100%** | | | | | | | 78% |
| 5 | 31 | 20 | 65% | 26 | 74% | | | | | | | 70% |
| 6 | 23 | 17 | 65% | 12 | 57% | 10 | 39% | 16 | 65% | | | 57% |
| 7 | 25 | 26 | 76% | 31 | 88% | 13 | 48% | | | | | 71% |
| **8** | **14** | **17** | **79%** | **23** | **93%** | | | | | | | 86% |
| 9 | 33 | 32 | 73% | 17 | 52% | 10 | 24% | 10 | 27% | 35 | 82% | 52% |
| 10 | 42 | 26 | 57% | 15 | 48% | 45 | 83% | 15 | 31% | | | 55% |
| 11 | 32 | 25 | 72% | 28 | 78% | 32 | 78% | 15 | 47% | | | 69% |
| **12** | **13** | **12** | **92%** | 13 | 85% | 11 | 77% | | | | | 85% |
| 13 | 12 | 14 | 75% | 16 | 75% | | | | | | | 75% |
| 14 | 23 | 11 | 39% | 13 | 61% | 15 | 57% | 12 | 48% | | | 51% |
| **15** | **17** | **27** | **94%** | 17 | 82% | | | | | | | 88% |
| 16 | 35 | 19 | 54% | 22 | 60% | 30 | 63% | | | | | 59% |
| **17** | **10** | **10** | **90%** | 11 | 80% | | | | | | | 85% |
| **18** | **15** | **14** | **73%** | **10** | **60%** | **22** | **93%** | | | | | 75% |
| 19 | 25 | 13 | 44% | 11 | 36% | 15 | 40% | 23 | 60% | | | 45% |
| **20** | **12** | **15** | **100%** | **16** | **92%** | | | | | | | 96% |
| 21 | 19 | 18 | 84% | 18 | 74% | | | | | | | 79% |
| 22 | 26 | 22 | 77% | 23 | 77% | 10 | 31% | | | | | 62% |
| 23 | 34 | 33 | 71% | 49 | 82% | 10 | 29% | | | | | 61% |
| 24 | 46 | 30 | 57% | 64 | 89% | 12 | 26% | 15 | 28% | | | 50% |

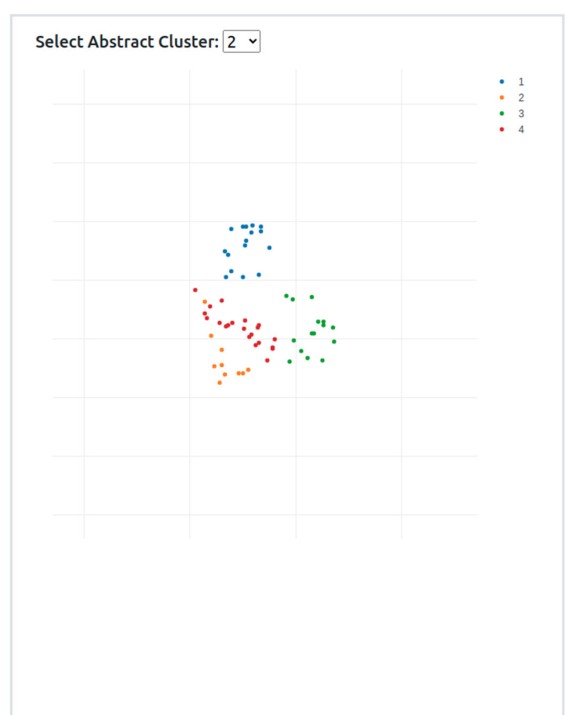
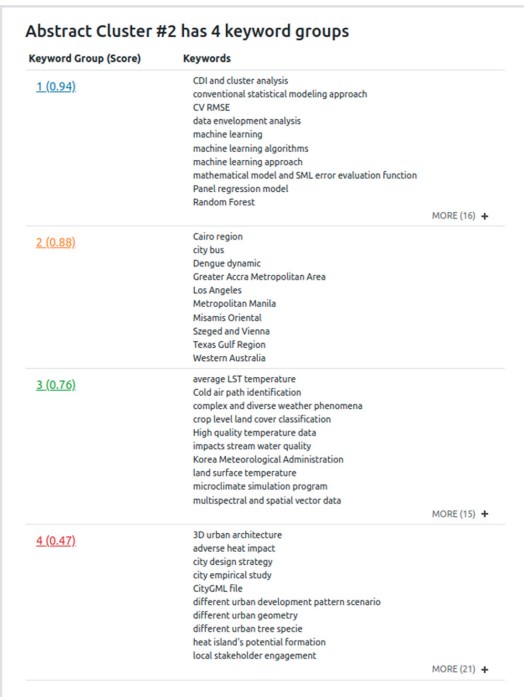

(**a**) The keyword groups of the abstract cluster #2. Clicking on a keyword group number on the right panel displays all keywords in (**b**).

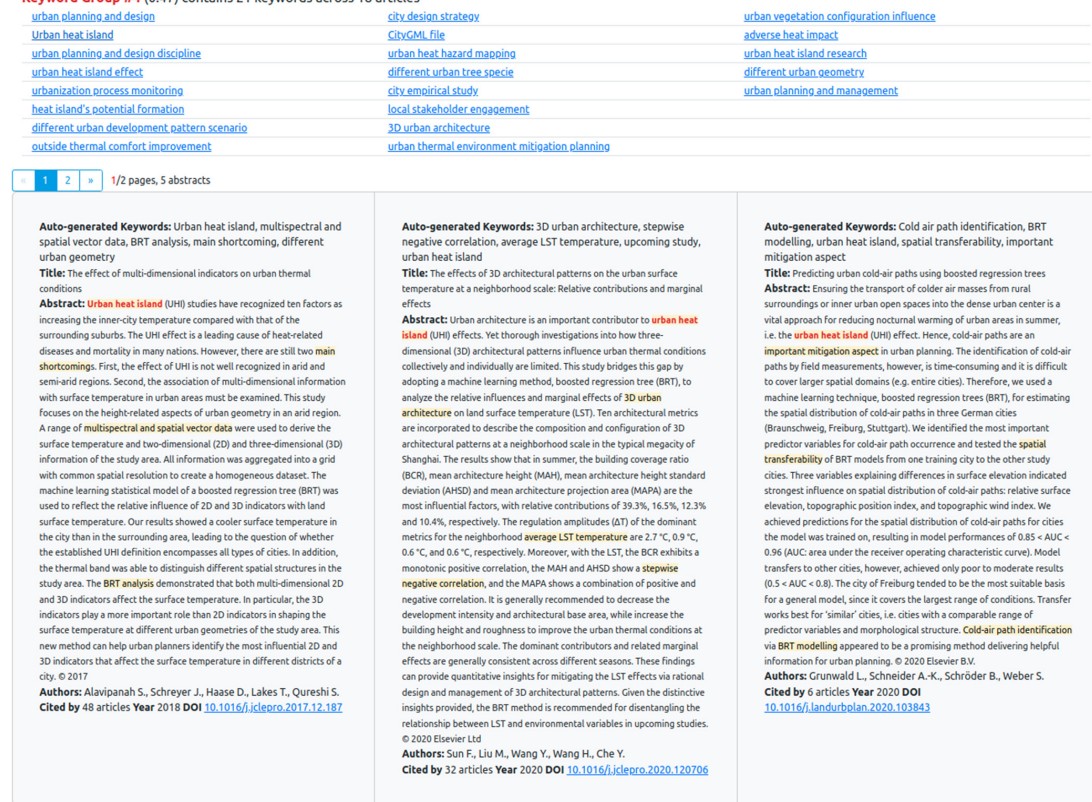

(**b**) The keywords in the keyword group #4. Clicking a keyword (*urban heat island*) on the top panel displays the abstracts containing the keyword in the bottom panel.

**Figure 7.** Keyword group visualization. By selecting the cluster #2, (**a**) displays a scatter chart of

four keyword groups in blue, orange, green, and red in the left panel and the keyword examples of each keyword group in the right panel. (**b**) displays the keyword group #4, including the keywords in the top panel and abstracts in the bottom panel.

## 5. Discussion

It should be noted that there are a number of limitations to our approach. For instance, we can only work on short texts because of the input size restriction of the GPT-3 similarity embedding models. At the moment, the maximum length of input text for the GPT-3 embedding models is 2048 tokens, which is sufficient for publication abstracts but not for the entire paper. The main factor that affects the accuracy of the document clustering results is the quality of the dense representation of the semantic meaning of the documents. Our approach relied on pre-trained embedding models that are typically trained on general purpose text data such as Wikipedia articles, books, or web pages. Some domain-specific models such as BioBERT [58] that fine-tune the BERT model on a small in-domain corpus such as scientific publications in a certain area are proposed and have provided significant improvement of domain-specific tasks. However, training or fine-tuning a GPT-3 similarity embedding model for this study is beyond our capacity.

Another issue faced during this study was how to handle outliers. The occurrences of outliers among the abstracts are unavoidable as they are retrieved from the Scopus database using two general terms: *machine learning* and *urban study* without manual filtering. This is evidenced by the fact that about 48.3% (287/593) of the abstracts were marked as outliers during the first clustering iteration (Table 2 in Section 4.1). One of the options for dealing with outliers is to discard them, which means abandoning almost half of our dataset. Instead, we chose to feed them as one cluster to the second iteration because some of the outliers are the data points falling on cluster borders. The results in Table 2 show that eight clusters (#12, #13, #14, #15, #17, #18, #19, #20) with reasonable silhouette scores (above 0.5) are generated in the subsequent iterations from outliers identified in the first iteration, which proves that our iterative clustering approach is practical and useful.

In addition, we used the unsupervised silhouette measure to evaluate the clustering results as it does not require training on ground-truth data to learn the formation of a cluster. This measure uses the distance metric to evaluate the clustering quality, whilst the HDBSCAN clustering algorithm we used is density-based. Having different metrics is common in clustering algorithms and evaluations [59] because the evaluation should have its own standard and have no preference for any algorithm. However, it would be interesting to experiment with the density-based silhouette score such as the one proposed by Menardi (2011) [60], which may help estimate the inconsistency in the clustering results caused by the use of different measures.

Our keywords extraction approach, inspired by BERT-Topic [56] and KeyBERT [57], utilizes the new development of the GPT-3 similarity embeddings and modern document clustering techniques (HDBSCAN). The quality of extracted keywords is determined by two factors: the pre-trained embedding models that are used to convert keywords to embeddings that represent the semantic meaning of the text and the selection criteria. The explanation of the keyword grouping can be demonstrated with our visualization. For example, in Figure 6a, two keywords *city bus* and *Dengue dynamic* in group 2 are distinct from the others (such as the *Cairo region*, *Vienna*, and *Los Angeles*). Through our keyword grouping visualization, we can retrieve the abstracts that these keywords come from and examine the sentences where these keywords appear to help understand the contexts of the keywords in the texts.

For the second factor, we used the Maximal Marginal Relevance (MMR) algorithm coupled with GPT-3 embedding to rank and select five keywords in each document. MMR's diversity parameter, with a value between 0 and 1, controls the relevance (similarities to the document) and the keyword diversity (similarities among keywords), and the diversity of 0.5 places equal importance on relevance and diversity. We investigated the effects of different diversity values. When the value is close to 0, more overlapping keywords

are produced, for example, *urban heat island research*, *urban heat island study*, and *urban heat island phenomenon*, as shown in Figure 4 in Section 3.2. On the other hand, when the value is close to 1, diverse and general keywords emerge such as *determinant variable*, *main contribution*, and *planning practice*. We decided to use a value of 0.5 as suggested by other similar studies [61–63]. However, the diversity parameter can be adjusted to meet the needs of specific applications [64].

There are some ways to further improve keyword extraction, which have not been explored by this study. Semi-supervised learning, unlike the supervised approaches that require labeled datasets for training, benefits from the portion of labeled data to improve its accuracy. The study by Wong et al. (2008) [65] showed that semi-supervised learning trained on both labeled and unlabeled data achieves a competitive performance compared to supervised learning. However, there are only a few semi-supervised solutions for keyword extraction at the moment [39]. Another way is to filter unwanted keywords. We note that some of our keywords are place names such as *Los Angeles*, *Manila*, and *Texas Gulf Region* in group 2 in Figure 6a. They were extracted because they occur frequently in the abstracts. In most cases, urban place names indicate the location a study takes place; however, they can be filtered out using named entity recognition techniques.

Finally, the evaluation of keyword extraction results is difficult and time-consuming because it typically requires human intervention to reach an agreement [66]. One is that domain experts use human-labeled datasets to judge the keyness of a word. Because no domain experts were available in this study, we treat the keyword grouping as the evaluation task with the silhouette score as the metric and present the keyword groups whose silhouette scores are positive. We tried topic coherence metrics such as UCI measure [67] or UMass measure [68] on the keyword grouping results, and found that these measures give different results because they use different metrics, e.g., UMass relies on word co-occurrences [1,69,70] and our approach on semantic similarity. We also attempted to use the context vector-based topic coherence metric proposed by Aletras and Stevenson (2013) [71] and the results are aligned with the silhouette scores, where the keyword groups with negative silhouette scores receive low scores.

## 6. Conclusions

This paper presents a study that investigates methods for extracting keywords from a collection of publications to help researchers gain an understanding of the current topics and themes in a research area. In doing so, a total of 593 abstracts of scientific publications in the fields of machine learning and urban study were used to demonstrate our approach. As such, we experimented with the GPT-3 similarity embedding models to represent the semantic meanings of abstracts and developed an iterative clustering method using HDBSCAN, together with the silhouette score, to divide the abstracts into a set of small clusters. Keywords, made up of two- to five-word compound nouns, were extracted from a cluster and grouped using the same method (HDBSCAN with the silhouette score) to represent the topics. Keywords were selected and ranked using a combination of the semantic similarity metric and maximum marginal relevance ranking algorithm to avoid over-presentation of similar words. The visualization interface allows researchers to inspect the clustering of abstracts and to examine the common topics in each cluster through keyword groups.

The construction of language models that accurately capture the semantic meaning of a text is an area of active ongoing research in natural language processing. Our approach will continuously benefit from the new advancement of language models in terms of achieving better accuracy. We envisage that such techniques have potential to be integrated into the existing bibliometric analysis tools for document clustering and keyword generation and visualization. Abstract and citation databases such as Scopus can extend the searching capabilities by incorporating auto-generated keywords into the human-produced keyword databases when building search indexes, therefore enhancing the user experience.

**Supplementary Materials:** The following are available online at https://github.com/samminweng/ AionUrbanStudies (accessed on 1 January 2022). Computer code and software was released and deposited in the public GitHub repository.

**Author Contributions:** Conceptualization, S.W., M.-H.W. and M.D.; methodology, S.W. and M.-H.W.; software, M.-H.W.; validation, S.W. and M.D; formal analysis, M.-H.W.; investigation, M.-H.W.; resources, M.-H.W.; data curation, M.-H.W.; writing—original draft preparation, S.W. and M.-H.W.; writing—review and editing, S.W., M.-H.W. and M.D.; visualization, M.-H.W.; supervision, S.W. and M.D.; project administration, M.D. and S.W.; funding acquisition, M.D. All authors have read and agreed to the published version of the manuscript.

**Funding:** This research received no external funding.

**Institutional Review Board Statement:** Not applicable.

**Informed Consent Statement:** Not applicable.

**Data Availability Statement:** Data are available in a publicly accessible repository. The data presented in this study are openly available in FigShare at (10.6084/m9.figshare.21092104.v1), reference number (Weng, Min-Hsien (2022): AIonUrbanStudies. figshare. Software. https://doi.org/10.6084/ m9.figshare.21092104.v1, accessed on 19 September 2022).

**Conflicts of Interest:** The authors declare no conflict of interest.

## Appendix A. Software Implementation

This project uses the Python and JavaScript programming languages with the open-source software libraries listed in Table A1.

**Table A1.** A list of software libraries used in our project.

| Software Libraries | Purposes | Phases |
|---|---|---|
| OpenAI API service [1] and pretrained GPT-3 model [2] (text-similarity-curie-001) | Abstract text and keyword vectorization | Phase 1 and Phase 2 |
| HDBSCAN clustering library [3] (v0.8.27) and UMAP dimension reduction library [4] (v0.5.1) | Abstract clustering and keyword grouping | Phase 1 and Phase 3 |
| Stanford CoreNLP [5] (v4.4.0) | POS tagging | Phase 2 |
| Plotly.js [6] (v2.11.1) | Scatter dot chart | Visualization |
| CSS Bootstrap [7] and Pagination.js [8] | List view | Visualization |

[1] OpenAI API service: https://openai.com/api/ (accessed on 1 January 2022). [2] Pretrained GPT-3 model: https://beta.openai.com/docs/models/gpt-3 (accessed on 1 January 2022). [3] HDBSCAN library (v0.8.27): https://hdbscan.readthedocs.io/en/latest/ (accessed on 1 January 2022). [4] UMap library (v0.5.1): https://github.com/lmcinnes/umap (accessed on 1 January 2022). [5] Stanford CoreNLP toolkit (v4.4.0): https://stanfordnlp.github.io/CoreNLP/ (accessed on 1 January 2022). [6] Plotly.js library (v2.11.1): https://plotly.com/javascript/ (accessed on 1 January 2022). [7] CSS Bootstrap library (v5.2.0): https://getbootstrap.com/ (accessed on 1 January 2022). [8] Pagination.js library (v2.1.5): http://pagination.js.org/ (accessed on 1 January 2022).

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
