# Peer review of "Identification and Visualization of Key Topics in Scientific Publications with Transformer-Based Language Models and Document Clustering Methods"

_applsci, doi:10.3390/app122111220_

Round 1
Reviewer 1 Report
This paper has been written logically, but there are lots of problems as follows:
(1) In the introduction section, the authors did not explain the contribution and the differences between your work and the related work.
(2) What are the superior ones compared with others? The authors should explain the excellence of your method in the introduction section.
(3) The materials and methods are clear and fully explained, but there is not enough evaluation. So, the evaluation matrix and performance measures should be described. Also, the results have to be validated.
(4) Experimental results should not only be described in detail but should also be compared with related or similar work. Also, describe your comparative analysis with others.
Reviewer 2 Report
This manuscript proposed the unsupervised detection approach for identifying and visualizing the key topics in scientific publications. This study is important for addressing the challenge of identifying suitable and related research topics in a diverse collection of publications. The core idea seems interesting, but the paper should be improved in some regards:
Title: The title should be improved to reflect the whole manuscript. Is it more about bibliometric analysis? Or is it more on unsupervised detection approach / clustering?
Abstract: What is the main motivation of this study? Various researchers have studied the unsupervised topic detection approach. What will the gap be addressed here? What is the main contribution? Faster detection approach? Please justify the finding. For abbreviation, please give the full name for the first time (e.g., GPT, HDBSCAN). Preferably not to use ‘We’ in the manuscript.
Keywords: Please add in other suitable keywords. Please remove ‘(List three…)’.
Contents: Please check the caption of the figures (e.g., Figure 5, Figure 7). Should describe more about the tables & figures rather than just reporting the results. Should explain the methodology in more detail on procedures especially on identification and visualization. Any time computation involves performing the procedures (e.g., the whole three phases)? It would be better to see more implication and justification of the findings for all the analysis for several aspects. So, it is preferable to have one section for Discussion instead of combining the Discussion and Conclusion in one section. Should highlight the conclusion/significant contribution and impact of this study.
Reviewer 3 Report
This is a very well-written and interesting article discussing how the recent advancements in the field of natural language processing, in particular topic modelling, can be applied for bibliometric analysis. I am rather impressed by the quality of the argument and also the methodological thoroughness, starting from the detailed (but not overtly complicated) method description and ending with a very well-thought discussion of the limitations. I also would like to note the excellent use of visual which helps the reader a lot to digest the argument. Overall, it is one of those very rare cases, where I would consider recommending to accept paper as it stands, but I still would leave here two points which I think might be of interest for the reader to include.
The first point concerns the practical implications of the study. I think it would be interesting if the author(s) would add a couple of sentences (max one paragraph) on how they invision the practical use of the approach they developed. Is it envisioned as a possible integration to existing bibliometric platforms? Or a way to help researchers generate the set of keywords which actually would align with what the above-mentioned platforms use to search for the information? Will (shall?) the automated keyword assignment actually replace the author-based keyword assignment?
The second point is related to the first one. I adored the interface examples provided in the paper, where the auto-generated keywords are matched to actual abstracts. Might it also be an option to add keywords assigned by the authors to the articles, so the reader would have a comparison between human-generated and auto-generated keywords? There should not be an extensive comparison / judgemental argument about one approach being better than another, but I think offering such an insight will add to the paper.
Round 2
Reviewer 1 Report
The comments of the reviewer have not been fixed.
Reviewer 2 Report
The authors have addressed the comments and suggestions. The manuscript has improved. Still, I have some additional remarks:
1) I suggest to the authors to elaborate more on the results obtained on the procedures of how to produce the visualization (e.g., in Fig. 5).
2) Any justification for selecting the combination of terms 'machine learning' and 'urban study' in identifying the list of publication.
3) Line 90: New line for Section 2.
